# Impact of thermogenesis induced by chronic β3-adrenergic receptor agonist treatment on inflammatory and infectious response during bacteremia in mice

Patrick Munro[1], Samah Rekima[2], Agnès Loubat[2], Christophe Duranton[3], Didier F. Pisani[3☯*], Laurent Boyer[1☯*]

1 Université Côte d'Azur, Inserm, C3M, Nice, France, 2 Université Côte d'Azur, CNRS, Inserm, iBV, Nice, France, 3 Université Côte d'Azur, CNRS, LP2M, Nice, France

☯ These authors contributed equally to this work.
* didier.pisani@univ-cotedazur.fr (DFP); laurent.boyer@univ-cotedazur.fr (LB)

## Abstract

White adipocytes store energy differently than brown and brite adipocytes which dissipate energy under the form of heat. Studies have shown that adipocytes are able to respond to bacteria thanks to the presence of Toll-like receptors at their surface. Despite this, little is known about the involvement of each class of adipocytes in the infectious response. We treated mice for one week with a β3-adrenergic receptor agonist to induce activation of brown adipose tissue and brite adipocytes within white adipose tissue. Mice were then injected intraperitoneally with *E. coli* to generate acute infection. The metabolic, infectious and inflammatory parameters of the mice were analysed during 48 hours after infection. Our results shown that in response to bacteria, thermogenic activity promoted a discrete and local anti-inflammatory environment in white adipose tissue characterized by the increase of the IL-1RA secretion. More generally, activation of brown and brite adipocytes did not modify the host response to infection including no additive effect with fever and an equivalent bacteria clearance and inflammatory response. In conclusion, these results suggest an IL-1RA-mediated immunomodulatory activity of thermogenic adipocytes in response to acute bacterial infection and open a way to characterize their effect along more chronic infection as septicaemia.

## 1. Introduction

White adipocytes, main constituents of white adipose tissue (WAT) are specialized in the storage and release of energy (carbohydrates, lipids), while brown adipocytes dissipate this energy in the form of heat (thermogenesis) [1]. Brown adipocytes constitute the brown adipose tissue (BAT) but can also be found within the WAT. They are then called beige or brite adipocytes (for "brown in white") and have an increased thermogenesis capacity in response to prolonged exposure to cold, for instance [2]. In addition to their seminal functions, an essential role in

**Data Availability Statement:** All relevant data are within the manuscript and its Supporting information files.

**Funding:** DFP received a fellowshig from "Société Francophone du Diabète (SFD)/Pierre Fabre Médicament" 2017. https://www.sfdiabete.org/ The funders had no role in the design of the study; in the collection, analyses, or interpretation of data; in the writing of the manuscript; or in the decision to publish the results.

**Competing interests:** The authors have declared that no competing interests exist.

the innate immunity has recently emerged since the characterization of several functional Toll-like receptors (TLRs) in the membrane of white and brown adipocytes [3–5]. TLRs are transmembrane receptors that play a critical role in the recognition and defense against all types of infectious pathogens, acting in innate, non-specific immunity [6]. TLRs, referred as pattern recognition receptors (PRR), are evolutionarily conserved receptors able to detect a repertoire of pathogen-associated molecular patterns (PAMPS) including constituent of the microbial cell walls such as the lipolysaccharides (LPS) of gram negative bacteria [6,7].

It is well-known that adipocytes respond to LPS stimulation, especially along obesity. During obesity, gut microbiota alteration (dysbiosis) and intestinal barrier disruption lead to an increased permeability to the microbiota degradation products, especially LPS. Obese patient's plasma displays an increased level of endotoxins (endotoxaemia) carried to organs and tissues, inducing a local activation of TLR-4 displayed by adipocytes and adipose tissue resident macrophages [8]. In response, adipocytes and macrophages secrete inflammatory cytokines (TNFα and IL-1β) and chemokines (MCP-1 and CXCL10) which promote local inflammation and alter tissue homeostasis [9]. Interestingly, it has been shown that brown and white adipocytes in vitro respond to LPS stimulation by modulating differently macrophage inflammatory responses. Conversely to white adipocytes, brown adipocytes reduced macrophage IL-6 secretion and mRNA expression of several inflammatory markers in response to LPS [10]. In this way, we recently demonstrated that after recruitment and activation of brown and brite adipocytes, mice displayed a higher anti-inflammatory response when they are exposed to lipopolysaccharides derived from *E. coli* bacteria [11]. Especially, these mice secreted high level of interleukin-1 receptor antagonist (IL-1RA) known to counteract interleukin 1β (IL-1β) inflammatory action [11].

In addition to the inflammatory response, LPS alter adipocyte function *via* TLR-4 activation or through cytokines secreted by macrophages. As an example, LPS induce white adipocyte lipolysis to inhibit adipogenesis [12,13]. A main response of organism to counteract infection is fever. Adipose tissues appear to be involved in this mechanism since WAT secretes leptin, an adipokine essential to pyrexia in rodent by limiting heat loss at the tail level, and BAT make thermogenesis suspected for a long time to participate to pyrexia [14]. Nevertheless, recent evidences have demonstrated that acute (*in vitro*) and chronic (*in vivo*) exposure to LPS could inhibit brown adipocytes function [5,15].

WAT is linked to metabolic inflammation but also to local and systemic defence mechanisms against pathogens infection. Indeed, adipocytes are able to respond to bacteria proximity with secretion of antimicrobial peptides as cathelicidin, lipocalin and defensin. First described for dermal adipocytes in response to *Staphylococcus aureus* infection [16], this characteristic has been extended to adipocytes from intra-abdominal and subcutaneous WAT [17]. Moreover, WAT is able to respond to bacterial infection as previously shown in case of sepsis, a severe infectious situation where adipocytes secrete adipokines known to modulate inflammation [18]. In this situation of sepsis, it was demonstrated that brite adipocytes recruitment was stimulated [19], but might be related to an increased norepinephrine level known to induce the browning of WAT [20].

The antimicrobial response of brown adipocytes has not been yet described despite their ability to respond to endotoxins. Indeed, cold exposure in human, known to induce brown and brite adipocytes activity and recruitment, increases the secretion of inflammatory cytokines in response to LPS and of IL-1RA independently to LPS [21]. These results were in line with our previous observations where we demonstrate the same data in a mice model of recruitment and activation of brown and brite adipocytes followed by an exposition to LPS [11]. Despite it is well established that adipocytes are able to detect and respond to bacteria via

TLR-2 and TLR-4, the involvement of the different adipocytes populations in this response is still unknown.

Here, we investigated the *in vivo* impact of recruitment/activation of brown/brite adipocytes by β-adrenergic receptors agonists on inflammatory response against *E. coli* infection in mice. We demonstrated that thermogenesis activity did not modify the host response to infection, without additive effect on fever and with an equivalent inflammatory response.

## 2. Materials and methods

### 2.1. Reagents

Media and buffer solutions were purchased from Lonza (Ozyme, St-Quentin en Yvelines, France). Other reagents were from Sigma-Aldrich (Saint-Quentin Fallavier, France).

### 2.2. Animals

The experiments were conducted in accordance with the French and European regulations (2010/63/EU directive) for the care and use of research animals and were approved by the french national experimentation committee (Ministère de l'Enseignement Supérieur, de la Recherche et de l'inovation, N°: APAFIS#18322–2018121809427035_v2). 8-week-old Balb/c male mice from Janvier Laboratory (France) were maintained at housing temperature (22°C) and 12:12-hour light-dark cycles, with *ad libitum* access to food and water. General health and behavior of mice were monitored daily. None of the mice died or were euthanized before the end of the experiments. All the mice used in the experiments have been included in the final analysis.

Mice were daily treated with β3-adrenergic receptor agonist CL316,243 (1 mg/kg in saline solution, intraperitoneal injection) (Sigma-Aldrich) or with vehicle only. For infection, treatment with CL316,243 was stopped after 7 day, then mice were injected with *E. coli* (UTI89 strain, $1.71 \times 10^7$ CFU/mouse in phosphate-buffered saline solution (PBS), caudal intra-venous injection) and monitored for 48 hours before sacrifice [22]. At 0, 4, 24 and 48 hours after bacterial infection rectal temperature was recorded with a digital thermometer. For the determination of bacteremia, blood was collected from the tail vein at the indicated times post-infection, serially diluted in sterile PBS and plated on LB plates. The plates were incubated for 16 h at 37°C before counting colonies.

At the end of experiment, mice were sacrificed by cervical dislocation and blood, interscapular brown adipose tissue (iBAT), epididymal (eWAT) and inguinal subcutaneous (scWAT) white adipose tissues were immediately sampled and used for different analyses. General parameters, plasma analysis and expression of adipose tissue molecular markers were evaluated in all the mice (n = 8); secretion of tissue explant and histology analysis were evaluated using whole fat pad or lobe from half of the mice of each group (n = 4).

### 2.3. Cytokine and metabolic parameters quantification

For blood analysis, freshly prepared plasmas were diluted twice before analysis. For tissue analysis, freshly sampled WAT and BAT were washed in PBS, weighed and incubated in free DMEM (Dulbecco's modified Eagle medium) for 2 hours at 37°C. Media were kept for various analyses of secreted proteins.

Leptin and cytokines were assayed using "Mouse Leptin Kit" (Meso Scale Discovery, # K152BYC) and "mouse V-PLEX Proinflammatory Panel 1 Kit" (Meso Scale Discovery, # K15048D) respectively, on a QuickPlex SQ 120 apparatus (Meso Scale Discovery) following manufacturer's instructions. IL-1RA levels were assayed using mouse IL-1RA Elisa kit

(#EMIL1RN) from Thermo Fisher Scientific (Courtaboeuf, France). Glycerol and triglycerides determinations were performed using dedicated kit (Free Glycerol reagent and Triglyceride reagent, Sigma Aldrich).

## 2.4. Histology

Freshly sampled tissues were fixed in 4% paraformaldehyde overnight at RT and then paraffin-embedded. Embedded tissues were cut in 5 μm sections and dried overnight at 37˚C. All sections were then deparaffinized in xylene, rehydrated with alcohol, and washed in phosphate-buffered saline (PBS). For histological analysis, the sections were stained with haematoxylin-eosin and mounted in vectamount (Vecto laboratories). For immunohistochemical analysis, antigen retrieval was performed in low pH buffer in a de-cloaking chamber (Dako, S2367). The sections were then permeabilized in PBS with 0.2% Triton X-100 at room temperature for 10 min and blocked in the same buffer containing 3% BSA for 1 hour. The sections were incubated with rat anti-F4/80 antibody (Biorad, clone Cl:A3-1, dilution 1:100) overnight at 4˚C. Following a 1-hour incubation with A568-coupled anti-rabbit secondary antibodies, nuclear staining was performed with DAPI, and the sections were mounted in PermaFluor mounting Media (Thermofisher).

## 2.5. Isolation and analysis of RNA

Total RNA was extracted using a TRI-Reagent kit (Euromedex, Souffelweyersheim, France) according to the manufacturer's instructions. Tissues were homogenized in TRI-Reagent using a dispersing instrument (ULTRA TURRAX T25). Reverse transcription-polymerase chain reaction (RT-PCR) was performed using M-MLV-RT (Promega). SYBR qPCR premix Ex Taq II from Takara (Ozyme, France) was used for quantitative PCR (qPCR), and assays were run on a StepOne Plus ABI real-time PCR instrument (PerkinElmer Life and Analytical Sciences, Boston). The expression of selected genes was normalized to that of the 36B4 (RPLP0) housekeeping genes and then quantified using the comparative-ΔCt method. Primer sequences are displayed in Table 1.

## 2.6. Statistical analysis

Animal cohort sizes have been determined using G*Power [23]. Data were analyzed using GraphPad Prism 6 software and statistical differences between experimental groups assessed by Kruskal-Wallis multiple comparison test. Differences were considered statistically significant with $p < 0.05$. Data were displayed as scatter plot of independent values and group mean values ± SEM.

## 3. Results

### 3.1. Bacterial acute infection does not modify response of mice to CL316,243

Mice were treated daily for 1 week with the β3-adrenergic receptor agonist CL316,243 (1 mg/kg, intraperitoneal), or vehicle only (NaCl), in order to induce brown adipose tissue (BAT) activation as well as recruitment and activation of brite thermogenic adipocytes within white adipose tissue (WAT). As expected, at the end of the treatment, mice body weight was decreased and rectal temperature increased in CL316,243 group compared to NaCl group (Fig 1A) due to the thermogenic activity of brown and brite adipocytes. At the end of this treatment, half of the mice of each group were infected with *E. coli* and monitored 48 hours as previously described [22]. While bacteria induced a decrease of body weight in NaCl treated mice,

**Table 1. Primers sequences used for qPCR analysis.**

|  | Forward | Reverse |
|---|---|---|
| 36b4 (Rplp0) | TCCAGGCTTTGGGCATCA | CTTTATCAGCTGCACATCACTCAGA |
| Adiponectin | CTTTCCTGCCAGGGGTTC | GGAGAGAAAGGAGATGCAGGT |
| Cd11b (Itgam) | TGACCTGGCTTTAGACCCTG | ACCTCTGAGCATCCATAGCC |
| Cd19 (Leu-12) | TCCCTGGGTCCTATGGAAAT | CTGGTCCTGCCCAAGGTT |
| Mrc1 | TGGATGGATGGGAGCAAAGT | GCTGCTGTTATGTCTCTGGC |
| Perilipin 1 | AGCGTGGAGAGTAAGGATGTC | CTTCTGGAAGCACTCACAGG |
| Perilipin 5 | CGCTCCATGAGTCAAGCCA | CTCAGCTGCCAGGACTGCTA |
| Tcrβ | CTCCACCCAAGGTCTCCTTG | GTGGTCAGGGAAGAAGCCC |
| Ucp1 | CACCTTCCCGCTGGACACT | CCTGGCCTTCACCTTGGAT |

no additive effect was found in CL316,243 treated mice (Fig 1B). Differently, CL316,243 treatment decreased epididymal WAT (eWAT) mass while bacteria did not modify it significantly. Plasma analysis revealed that CL316,243 treated mice displayed higher glycerol level (Fig 1C) and lower triglycerides level (Fig 1D) compared to NaCl group and independently of bacteria treatment. These could be related to an increase fatty acid oxidation in link with thermogenesis. Interestingly, an additive effect was found for triglycerides plasma level in mice treated with CL316,243 and infected with bacteria.

While histological analysis of inter-scapular BAT (iBAT) performed 48 hours after the end of CL316,243 treatment, did not shown major differences between groups (Fig 1E), molecular analysis showed an increased Ucp1 and Perilipin 5 mRNA expressions after CL316,243 treatment and independently of bacteria injection, which is characteristic of activated BAT (Fig 2A). Differently, Perilipin 1 and adiponectin mRNA expression was not affected by treatments (Fig 2A). Sub-cutaneous WAT (scWAT), and for a lower extend eWAT, showed numerous multilocular adipocytes in CL316,243 treated mice, characteristic of brite adipocytes (Fig 1E). This was confirmed by the overexpression of Ucp1 and Perilipin 5 mRNA in scWAT (Fig 2B). Perilipin 1 and adiponectin mRNA expression in scWAT did not change with CL316,243 treatment (Fig 2B). Bacteria did not modify the frequency of brite adipocytes as well as iBAT morphology (Fig 1E) which was corroborated with brown/brite adipocyte markers molecular analysis (Fig 2).

## 3.2. Bacterial clearance during bacteremia and pyrexia are independent to leptin systemic level and BAT activation

Intraperitoneal injection of bacteria in mice induces in the first 4 hours a pyretic response which was maintained all along the two days of analysis (Fig 3A). Interestingly, even if rectal temperature was different before infection (Fig 1A), the fever measured 4 hours after bacterial infection was equivalent between NaCl- and CL316,243-treated mice. As expected, the bacteremia was decreased at 24 hours and completely resolved after 48 hours, demonstrating bacteria clearance by mice (Fig 3B) [22]. Activation of brown and brite adipocytes did not modify this response to infection (Fig 3B).

Leptin is considered as a key mediator of pyrexia especially in mice by limiting body heat loss. Interestingly, we measured a decreased leptinemia as well as leptin secretion by scWAT in mice infected with bacteria as well as in mice treated with CL316,243 (Fig 3C), demonstrating that pyrexia in response to bacteria and thermogenesis are independent to leptin action.

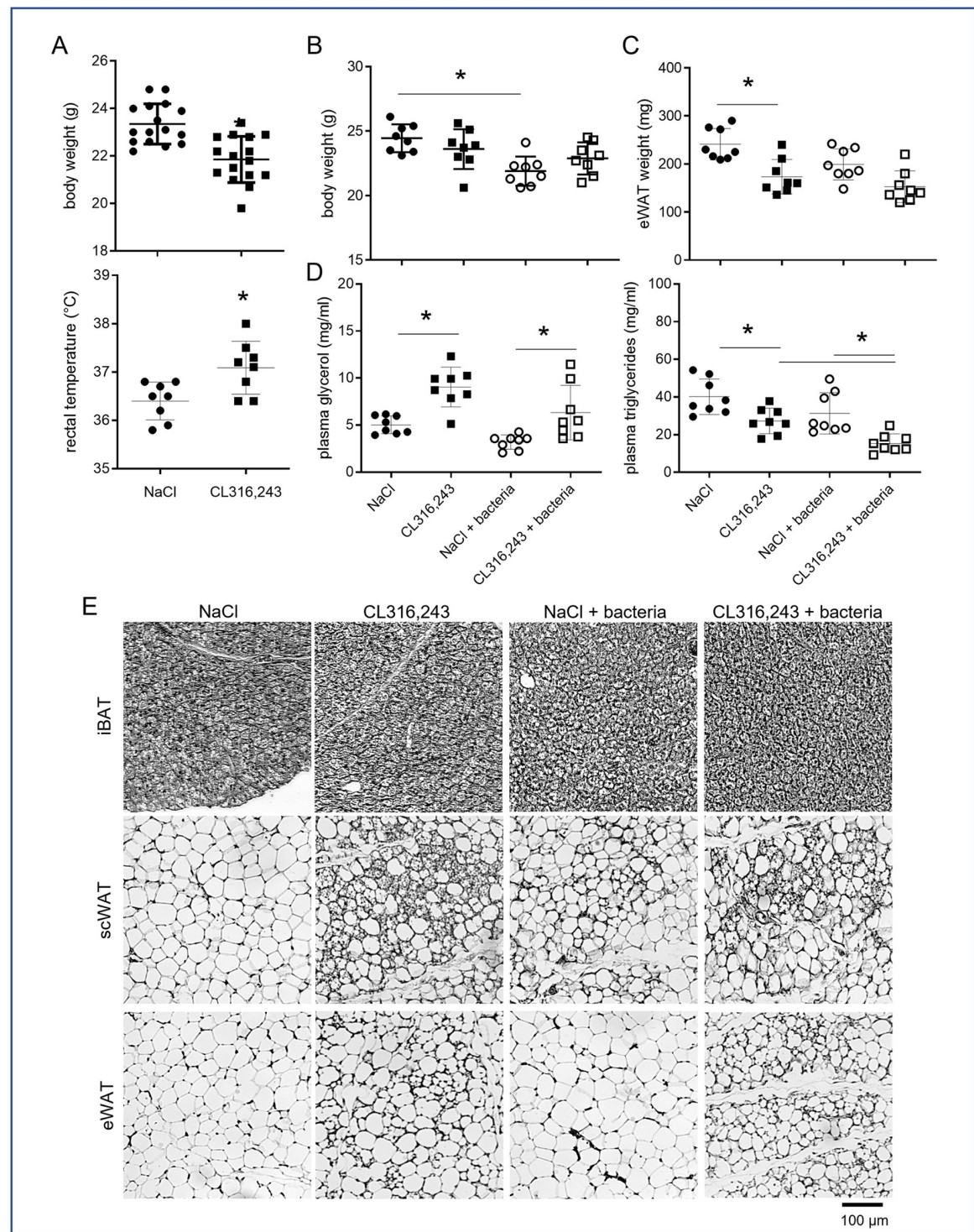

**Fig 1. Bacterial infection does not modify metabolic parameters and adipose tissue morphology.** Mice were analysed after 1 week of treatment with CL316,243 daily (1 mg/kg/day) or vehicle only (NaCl), and with or without *E. coli* infection for 48 hours. (A) Mouse body weights and rectal temperatures at the end of CL316,243 treatment. (B) Mouse body weights and (C) epididymal white adipose tissue (eWAT) weights at the end of infection. (D) Triglycerides and glycerol plasma levels at the end of experiment. (E) Representative picture of four haematoxylin-eosin staining of interscapular BAT (iBAT), sub-cutaneous (scWAT) and eWAT sections. The results are displayed as independent values (dots) and the mean ± SD. n = 16 (A) or 8 (B-D). * *p*<0.05.

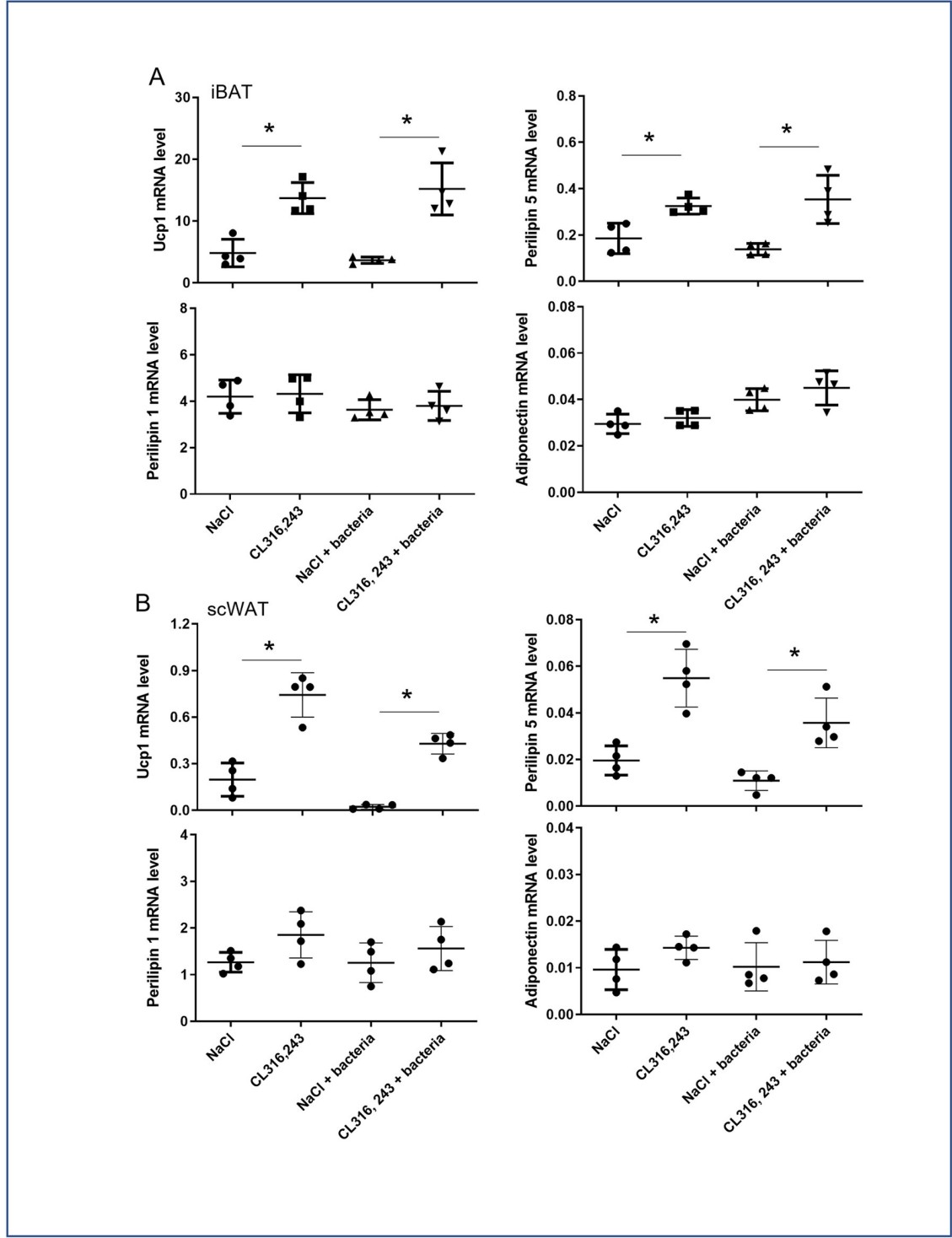

**Fig 2. Effect of CL316,243 and bacteria on mRNA of adipocyte markers.** mRNA expression of brown/brite (Ucp1 and perilipin 5) and general adipocyte (perilipin 1 and Adiponectin) markers was analysed by qPCR in iBAT and scWAT from mice that were treated for 1 week with CL316,243 daily (1 mg/kg/day) or vehicle only (NaCl), and then infected or not with *E. coli* for 48 hours. The results are displayed as independent values (dots) and the mean ± SD. n = 4. * $p<0.05$.

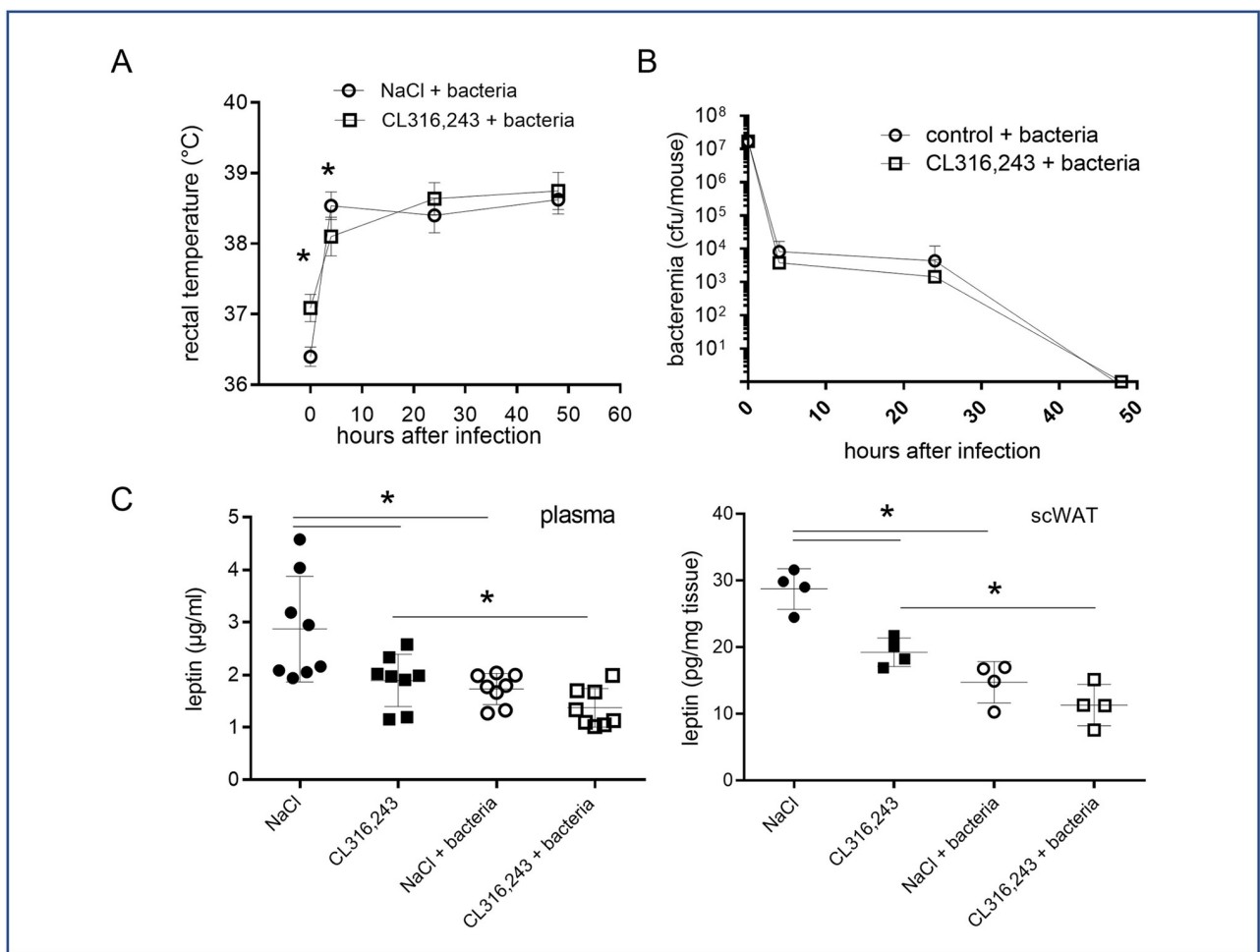

**Fig 3. Pyrexia, bacteremia and leptin secretion in response to infection.** (A) Rectal temperature and (B) bacteremia of mice treated or not with CL316,243 (1 mg/kg/day; 1 week) were monitored for 48 hours after infection by *E. coli*. (C) Mice plasma and scWAT explant leptin levels at the end of experiment. The results are displayed as independent values (dots) and the mean ± SD. n = 8 (mice and plasma) or 4 (scWAT explants). * $p < 0.05$.

### 3.3. Systemic and local inflammatory response to bacteremia

We found that CL316,243 preconditioning did not modify fever and bacteria clearance by mice. Thus, we analyzed cytokine profiling to determine if brown and brite adipocyte activations were modulating the immune response to bacterial infection. As expected, bacterial infection induced a systemic inflammatory response characterized by increased plasmatic levels of TNFα, IL-6, IL-12, IFNγ and KC/GRO (CXCL-1) compared to PBS-treated mice (Fig 4A). We observed that CL316,243 treatment did not modulate basal plasma cytokine levels as well as in response to bacteria, even if a slight decrease was found (Fig 4A). To note, the high variability in plasma quantity for each cytokine (Fig 4A) in response to bacteria could be related to the variability in bacteremia evolution we found between mice (Fig 3B).

Interestingly, in iBAT only IFNγ secretion increased in response to bacteria injection (Fig 4B), other cytokines were either unaffected (TNFα, IL-2) or decreased (IL-6, IL-12, KC/GRO) (Fig 4B). In scWAT, most cytokine secretions were unaffected except for IL-12 which was decreased (Fig 4C). In all case, CL316,243 did not modulate local cytokine secretions in response to infection (Fig 4B and 4C).

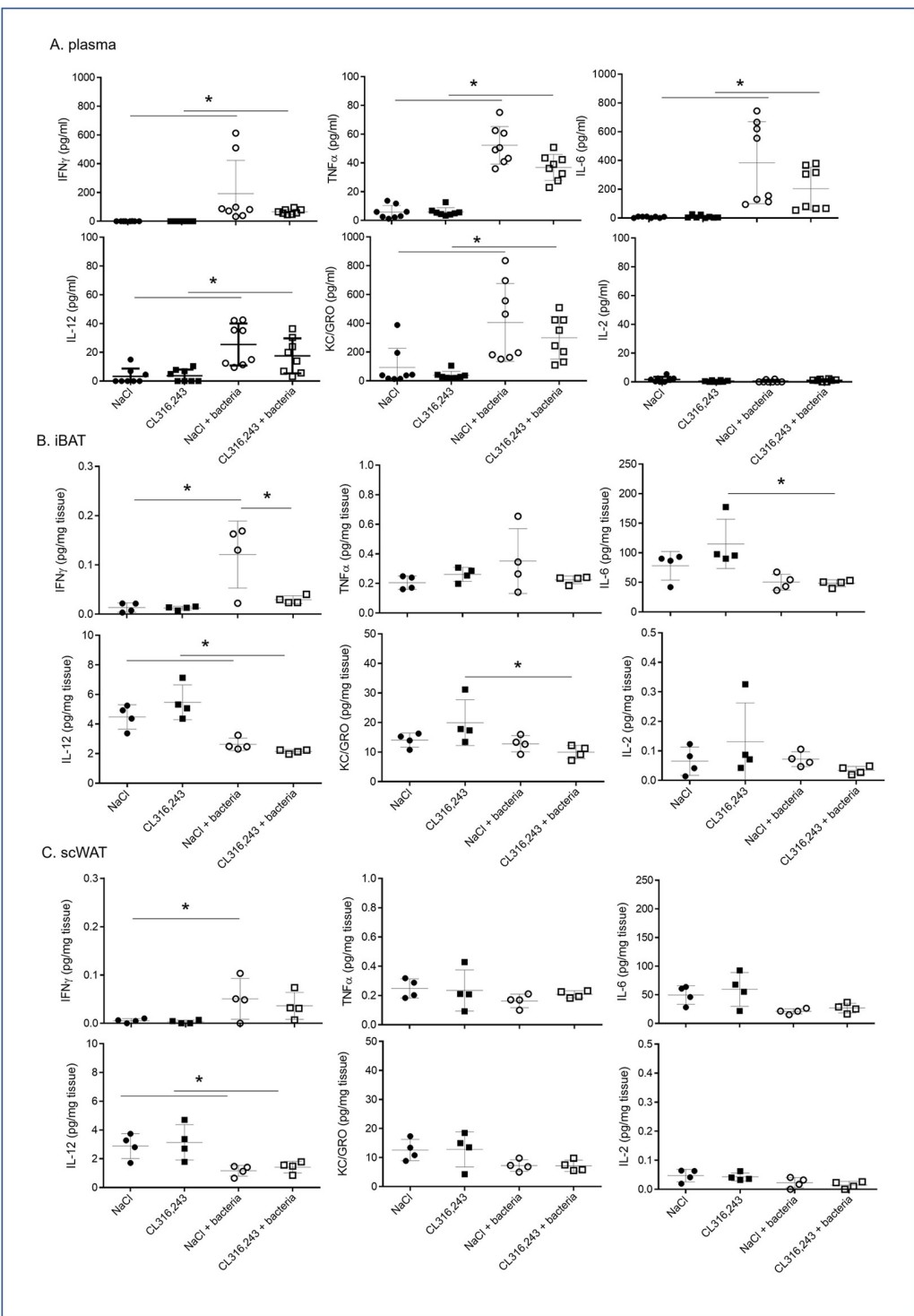

**Fig 4. Systemic and local inflammatory responses to infection.** IFNγ, TNFα, IL-6, IL-12-p70, KC/GRO (CXCL-1) and IL-2 levels were assessed in the plasma (A) or in the media of iBAT (B) and scWAT (C) explants from mice that were treated for 1 week with CL316,243 daily (1 mg/kg/day) or vehicle only (NaCl), and infected with *E. coli* for 48 hours. The results are displayed as independent values (dots) and the mean ± SD. n = 8 (plasma) or 4 (explants). * $p < 0.05$.

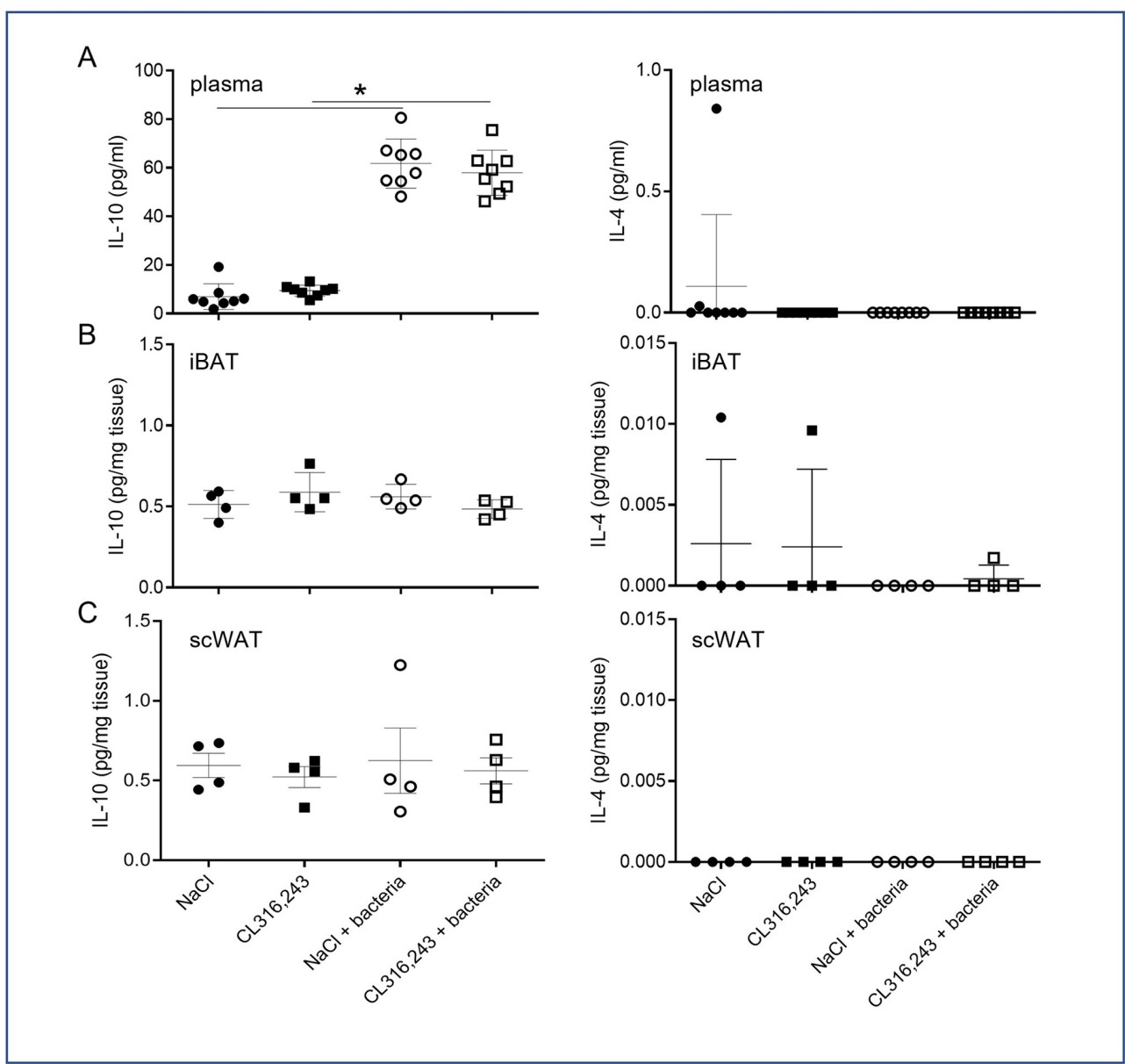

**Fig 5. Systemic and local anti-inflammatory response to bacteria.** IL-4 and IL-10 levels were assessed in plasma (A) and in the media of iBAT (B) and scWAT (C) explants from mice that were treated for 1 week with CL316,243 daily (1 mg/kg/day) or vehicle only (NaCl), and infected with *E. coli* for 48 hours. The results are displayed as independent values (dots) and the mean ± SD. n = 8 (plasma) or 4 (explant). * $p < 0.05$.

We further analyzed the impact of CL316,243 pretreatment on inflammatory resolving. To this aim we analyzed IL-4 and IL-10 levels in plasma and in secretory media of adipose tissue explant. Unfortunately, IL-4 was never detected and even if IL-10 level increased in plasma of infected mice, it was not modulated by CL316,243 treatment (Fig 5). As for pro-inflammatory cytokines, IL-10 secretion was unaffected by bacteria as well as CL316,243 treatment in iBAT and scWAT (Fig 5).

Next, we investigated the immune cells infiltration in the scWAT of infected mice. We found classic figures of inflammation in the scWAT of mice infected by bacteria. The immune cells infiltration in the scWAT was found to be similar in mice pretreated or not with

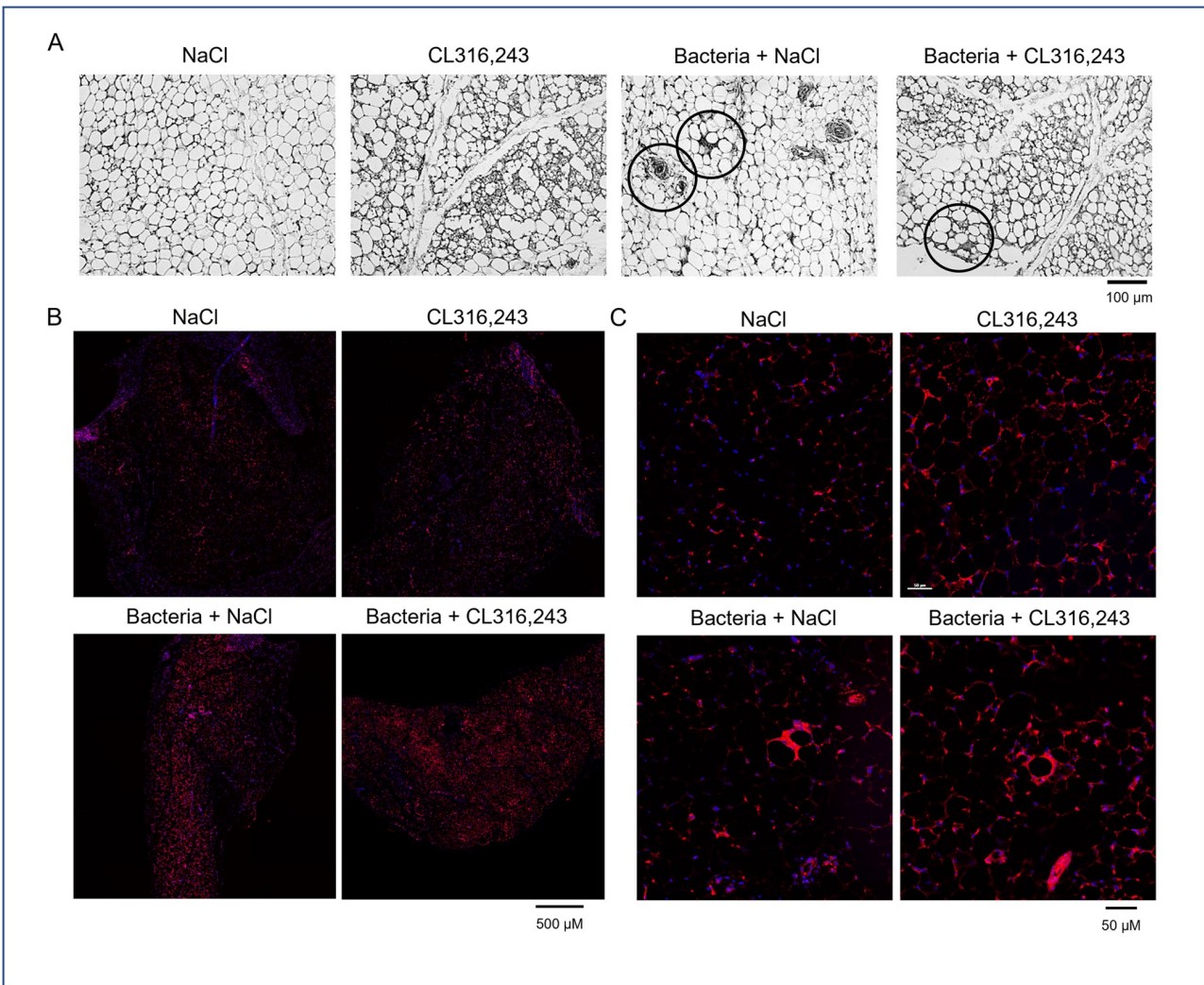

**Fig 6. Histological characterization of scWAT infiltration by immune cells.** (A) Representative pictures of four haematoxylin-eosin staining of sub-cutaneous white adipose (scWAT) tissue of mice treated or not with CL316,243 (1 mg/kg/day; 1 week) and infected 48 hours with *E. coli*. Typical figures of immune cell infiltration and crown structure are black circled. (B, C) Low and high magnification of F4/80 immunostaining (in red) in scWAT of the same mice. Scale bars are indicated.

CL316,243, including crown structure and small cell infiltration, potentially related to macrophage and monocyte or lymphocyte presence respectively (Fig 6A). This was confirmed by a F4/80 immunostaining clearly showing macrophage occurrence in adipose tissue (Fig 6B). To improve detection of immune cell content in whole adipose tissues, we have analyzed mRNA expression in iBAT and scWAT of general markers for B lymphocytes (Cd19), T lymphocytes (Tcrβ) and monocytes/macrophages (Cd11b), as well as expression of Mrc-1, a marker of anti-inflammatory macrophages. The results displayed in Fig 7 showed that only Cd11b increase with bacterial infection. Cd19, Tcrβ and Mrc-1 mRNA levels were equivalent between groups. These results corroborated histology results suggesting that only monocytes and inflammatory macrophages have infiltrated adipose tissue of mice in response to bacterial infection. To note, immune cells markers were unaffected by CL316,243 treatment (Fig 7).

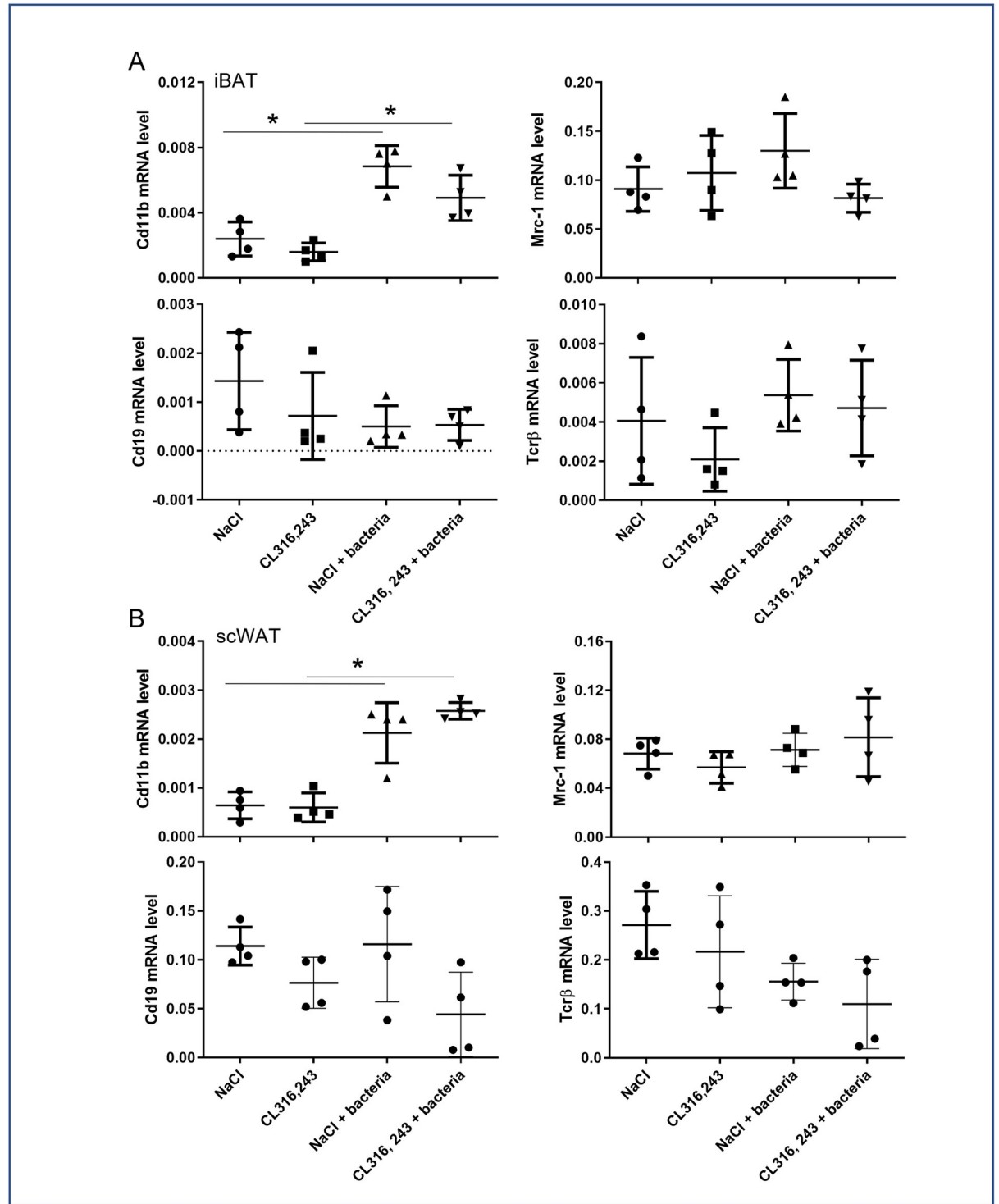

**Fig 7. Effect of CL316,243 and bacteria on mRNA of immune cell markers.** mRNA expressions of monocyte/macrophage (Cd11b), anti-inflammatory macrophage (Mrc-1), T (Tcrβ) and B (Cd19) lymphocytes were analysed by qPCR in iBAT and scWAT from mice that were treated for 1 week with CL316,243 daily (1 mg/kg/day) or vehicle only (NaCl), and then infected or not with *E. coli* for 48 hours. The results are displayed as independent values (dots) and the mean ± SD. n = 4. * $p < 0.05$.

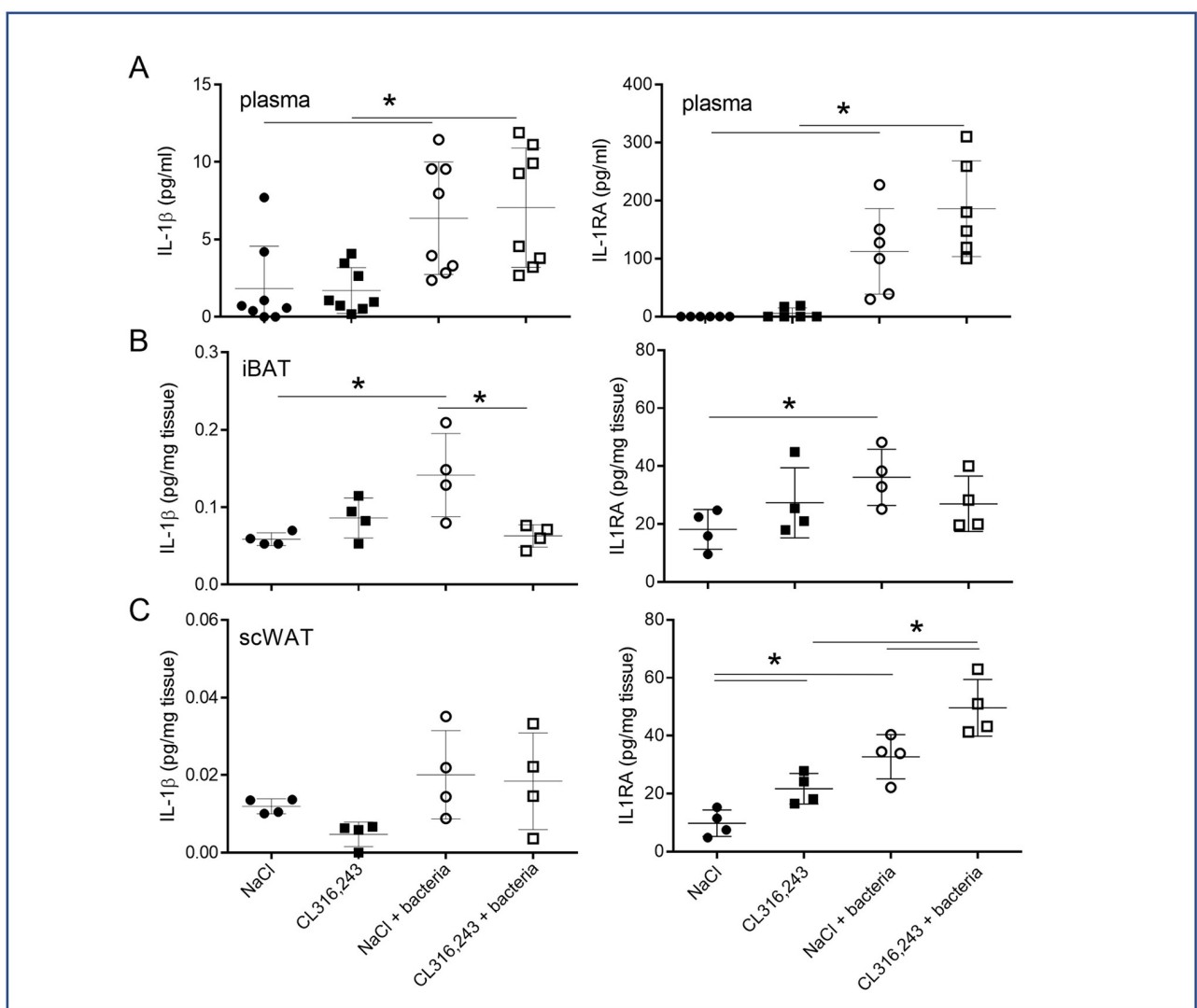

**Fig 8. IL-1 pathway response to bacterial infection.** IL-1β (A) and IL-1RA (B) protein levels were assessed in plasma (upper panel) and in the media of iBAT (middle panel) and scWAT (lower panel) explants from mice that were treated for 1 week with CL316,243 daily (1 mg/kg/day) or vehicle only (NaCl), and with or without *E. coli* infection for 48 hours. The results are displayed as independent values (dots) and the mean ± SD. n = 8 (plasma) or 4 (explant). * $p<0.05$.

### 3.4. Systemic and local IL-1β and IL-1RA level in response to bacteria

Among inflammatory cytokines, IL-1β is a main actor of pyrexia and anti-bacterial response and is highly secreted by the adipose tissue. Thus, in response to bacterial infection, we found the IL-1β and IL-1RA (IL-1 receptor antagonist) plasma levels increased concomitantly (Fig 8A). Pre-treatment of mice with CL316,243 did not modify this response (Fig 8A). Analysis of IL-1β and IL-1RA secretions in iBAT showed that both increased in infected mice and, interestingly, IL-1β secretion decreased in iBAT activated by CL316,243 treatment (Fig 8B). In contrast, IL-1RA level was equivalent in iBAT of infected mice treated or not with CL316,243 (Fig 8B). The IL-1β secretion by the scWAT was not significantly increased by bacteria infection but we measured an increased IL-1RA secretion. The CL316,243 treatment increased IL-1RA secretion with an additive effect to infection (Fig 8C).

## 4. Discussion

Adipose tissue is involved in local and systemic response to infection especially due to the expression of Toll-like receptors by adipocytes [3,4]. Indeed, sepsis as well as endotoxaemia directly activate inflammatory response in adipose tissue and alter its homeostasis [5,17,18]. Recent studies have shown that adipose tissue interacts with parasites or bacteria and proposed adipocytes as a potential reservoir [24,25]. In another hand, adipose tissue contains numerous immune cells such as macrophages, eosinophils or lymphocytes that can participate to the resolution of the infection [12]. Recently, it has been demonstrated that mice infected with *Yersinia pseudotuberculosis* accumulated memory T lymphocytes in adipose tissue, which are able to protect mice from a secondary infection [26].

While various types of adipose tissue exist [27], studies have focused only on white adipose tissue and white adipocytes. In this work, we focused on the impact of brite and brown adipocytes, both displaying a thermogenic function, on the local and systemic inflammation in response to bacterial infection. Indeed, as thermogenic adipocytes have been suspected to be involved in fever and anti-inflammatory response [1], we hypothesized that activating brite and brown adipocytes could lead to a better response against bacteria. Instead of the more physiologically cold exposition which activates numerous pathways in addition to thermogenesis, we have treated mice with a β3-adrenergic agonist leading to a more specific activation of thermogenic pathway. Our results have shown that active thermogenesis did not influence immune response to bacterial infection, including bacteria clearance and cytokines release. Interestingly, we did not find any additivity between pyrexia and thermogenesis. Indeed, mice displaying higher body temperature after CL316,243 treatment did not increase their body temperature after bacteria injection but maintained it to a similar level that the one measured for control infected mice. As bacteria did not modify BAT and scWAT phenotype and function, we can speculate that thermogenesis replaced in part classic pyretic response. In this line, leptin has been described as a major player in pyrexia in mice [28,29]. In our work, we corroborated a decrease in leptin level and secretion with an increase body temperature in response to infection, thus excluding a positive role of leptin in biological response. It would be interesting in the future to reproduce our work using *Ucp1*-knock out mice known to display lowered thermogenic activity, to clearly delineate the role of thermogenesis in pyretic response to bacterial infection.

Numerous studies have demonstrated an alteration of white and brown adipocytes function due to LPS exposition *in vivo* and *in vitro* [4,12,15,17,30]. In contrast to results obtained with LPS, *E. coli*-triggered acute infection did not seem to modify white and brown adipose tissue function, at least for lipolysis, lipogenesis, thermogenesis and morphology. Another difference exists between LPS and bacteria exposition, while LPS led to increase cytokine secretion in the adipose tissue [10,11,31], bacteria induced a limited reaction despite a significant increase in plasma cytokine levels as well as pro- and anti-inflammatory cytokines. These discrepancies between results found with LPS and *E. coli* bacteremia ask the question about the amount and the diffusion of *E. coli* LPS during bacteremia as well as physiological relevance of the LPS triggered sepsis model. Mice display high capacity to eliminate bacteria after 48 to 72 hours [32], a characteristic retrieved in our work, and we can speculate that this fast clearance could preclude the bacteria to target peripheric tissue. This could explain the low inflammatory response found in adipose tissue in term of cytokine release, and the infrequent immune cells infiltration or crown structures. This is a situation completely different from sepsis where bacterial infection was chronic and adipose tissue displayed a sustain inflammatory phenotype [17,33]. Thus, our study supports the hypothesis of an innate immune function of the adipose tissue in chronic rather than in acute infections.

CL316,243 treatment affected slowly the inflammatory response to bacteria of our mice. In control condition, chronic β-adrenergic treatment did not modify cytokine secretion, as well as immune cell content and macrophage phenotype as suggested by molecular analysis. In infected mice, plasma levels of IL-6, IL-12, TNFα, IFNγ and KC/GRO increased and were unaffected by pre-treatment with CL316,243. In adipose tissues, only IFNγ secretion increased after infection and this was blunted by CL316,243 treatment only in BAT. The same profiles were found for IL-1β. IL-6 and KC/GRO secretions by BAT were not affected in infected control mice but decreased in CL316,243 infected mice. Thus, activation of BAT thermogenesis seemed to lead to a lower inflammatory response against bacteria. In addition to these pro-inflammatory cytokines, IL-12 has a central role in the immune response to bacteria [34] and results obtained for its secretion are questionable. Indeed, while IL-12 increased significantly in plasma of our mice as expected, we found a decrease in IL-12 after bacterial infection in adipose tissues independently of CL316,243 treatment.

Anti-inflammatory cytokines secretion by adipose tissue were unaffected by CL316,243 and during bacteraemia. Nevertheless, we showed that bacterial infection increased IL-1RA levels in plasma and BAT independently of CL316,243 treatment and, in contrast, scWAT levels of IL-1RA increased in response to infection and to CL316,243 with an additive effect. IL-1RA, which is highly produced by adipose tissue [35], antagonizes IL-1β cell response by competing for binding to the IL-1 receptor [36]. These results demonstrated that WAT containing brite adipocytes secreted more IL-1RA than WAT and could play an important anti-inflammatory role to control the inflammatory response during bacteremia.

Altogether these results demonstrated that adipose tissues displayed a discreet inflammatory response to acute bacteria exposition during bacteremia with a bare cytokine release and immune cells infiltration. As our study corresponds to an end-point analysis after acute bacteria treatment, we cannot exclude that kinetic analysis with earlier points after infection, as well as multiplicity of infection can modify our conclusions. Nevertheless, it could be interesting to characterize the role of brite adipocytes in case of local or more sustained infections as sepsis, as they released high quantity of IL-1RA in response to bacteria exposition, a cytokine known to counteract IL-1β dependant inflammation.

## Supporting information

**S1 Dataset.**
(XLSX)

## Acknowledgments

The authors greatly acknowledge the C3M Animal core facility and the IBV histology and cytometry platforms.

## Author Contributions

**Conceptualization:** Didier F. Pisani, Laurent Boyer.

**Data curation:** Patrick Munro, Samah Rekima, Agnès Loubat, Christophe Duranton, Didier F. Pisani.

**Formal analysis:** Patrick Munro, Samah Rekima, Agnès Loubat, Christophe Duranton, Didier F. Pisani, Laurent Boyer.

**Funding acquisition:** Didier F. Pisani.

**Investigation:** Patrick Munro, Samah Rekima, Didier F. Pisani.

**Methodology:** Patrick Munro, Samah Rekima, Didier F. Pisani, Laurent Boyer.

**Supervision:** Didier F. Pisani, Laurent Boyer.

**Validation:** Didier F. Pisani, Laurent Boyer.

**Writing – original draft:** Didier F. Pisani, Laurent Boyer.

**Writing – review & editing:** Patrick Munro, Didier F. Pisani, Laurent Boyer.

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
