## [Decision Letter · Decision Letter 0]

28 Jun 2021

PONE-D-21-05608

Impact of thermogenesis induced by chronic β3-adrenergic receptor agonist treatment on inflammatory and infectious response during bacteraemia in mice

PLOS ONE

Dear Dr. Pisani,

Thank you for submitting your manuscript to PLOS ONE. After careful consideration, we feel that it has merit but does not fully meet PLOS ONE’s publication criteria as it currently stands. Therefore, we invite you to submit a revised version of the manuscript that addresses the points raised during the review process.

We look forward to receiving your revised manuscript.

Kind regards,

Fulvio D'Acquisto, PhD

Academic Editor

PLOS ONE

Journal Requirements:

2. As part of your revisions, we request additional details pertaining to research animal animal management, health and well-being. Please address the following items: (1)  monitoring parameters (the physical and behavioral criteria used to evaluate health and well-being); (2) unanticipated adverse events (animal sickness stemming from the experiment); (3) rate of moribundity and/or mortality (number of animals that became ill or died at any point during the study; (4) details about humane endpoints; (5) method of euthanasia; (6) any additional steps taken to minimize potential pain and distress."

3. Thank you for including your ethics statement:  "The experiments were conducted in accordance with the French and European regulations (2010/63/EU directive) for the care and use of research animals and were approved by national experimentation committees (MESR No.: APAFIS#18322-2018121809427035 v2). Mice have been sacrified by cervical dislocation.".  

Please amend your current ethics statement to include the full name of the ethics committee that approved your specific study.

For additional information about PLOS ONE submissions requirements for ethics oversight of animal work, please refer to http://journals.plos.org/plosone/s/submission-guidelines#loc-animal-research 

“This research was funded by INSERM, Region PACA and Société Francophone du Diabète (SFD)/Pierre Fabre Médicament 2017. “

“DFP received a fellowshig from "Société Francophone du

Diabète (SFD)/Pierre Fabre Médicament" 2017.

https://www.sfdiabete.org/”

5. In your Data Availability statement, you have not specified where the minimal data set underlying the results described in your manuscript can be found. PLOS defines a study's minimal data set as the underlying data used to reach the conclusions drawn in the manuscript and any additional data required to replicate the reported study findings in their entirety. All PLOS journals require that the minimal data set be made fully available. For more information about our data policy, please see Upon" ext-link-type="uri" xlink:type="simple">http://journals.plos.org/plosone/s/data-availability."Upon re-submitting your revised manuscript, please upload your study’s minimal underlying data set as either Supporting Information files or to a stable, public repository and include the relevant URLs, DOIs, or accession numbers within your revised cover letter. For a list of acceptable repositories, please see http://journals.plos.org/plosone/s/data-availability#loc-recommended-repositories. Any potentially identifying patient information must be fully anonymized.

Important: If there are ethical or legal restrictions to sharing your data publicly, please explain these restrictions in detail. Please see our guidelines for more information on what we consider unacceptable restrictions to publicly sharing data:

http://journals.plos.org/plosone/s/data-availability#loc-unacceptable-data-access-restrictions. Note that it is not acceptable for the authors to be the sole named individuals responsible for ensuring data access.

Reviewers' comments:

Reviewer's Responses to Questions

**Comments to the Author**

1. Is the manuscript technically sound, and do the data support the conclusions?

Reviewer #1: Yes

Reviewer #2: Yes

Reviewer #3: Yes

2. Has the statistical analysis been performed appropriately and rigorously? 

Reviewer #1: Yes

Reviewer #2: Yes

Reviewer #3: Yes

3. Have the authors made all data underlying the findings in their manuscript fully available?

Reviewer #1: Yes

Reviewer #2: Yes

Reviewer #3: Yes

4. Is the manuscript presented in an intelligible fashion and written in standard English?

Reviewer #1: Yes

Reviewer #2: No

Reviewer #3: Yes

5. Review Comments to the Author

Reviewer #1: General information:

The article is well written, the authors analyze the impact of thermogenesis induced by chronic β3-adrenergic receptor agonist treatment on inflammatory and infectious response during bacteremia in mice. In particular, there is a strong focus on the role of pro and anti-inflammatory cytokines. The authors communicate that brown and beige adipocytes activation/recruitment have a very limited impact on the host response to infection.

Strengths:

The whole article is easy to read, and the figures are clear. In addition, the objective of the study as well as the limitations of current knowledge are well highlighted. Finally results and discussion are well organized and present the main finding of the study.

Limitations:

The models used are unfortunately limited, activation of brown adipocytes and recruitment of beige adipocytes is limited to utilization of a single adrenergic agonist and the infection is limited to the utilization of a single dose of a single bacterial strain. In addition to CL316.243 utilization, authors could expose mice to cold which is the main brown/beige fat activator. In addition to this gain of function, a loss of function approach based on utilization of knock out mice (UCP1-KO for example) mice could have been considered.

Minor comments:

• In the second paragraph of the introduction the authors discuss a different response of beige and brown adipocytes after in vitro stimulation with LPS: «Interestingly, it has been shown that brown and white adipocytes respond differently to in vitro LPS stimulation and mediate different inflammatory responses (10)». A sentence explaining what the differences are would be appreciated.

• In the second paragraph of the introduction, it is mentioned that after activation of brown and recruitment of beige adipocytes, mice infected with LPS exhibit a higher pro-inflammatory response characterized by increased secretion of IL-1RA «We recently demonstrated that after recruitment and activation of brown and brite adipocytes, mice displayed a higher pro-inflammatory response when they are exposed to lipopolysaccharides derived from E. coli bacteria (11). Especially, these mice secreted high level of interleukin-1 receptor antagonist (IL-1RA) known to counteract interleukin 1β (IL-1β) inflammatory action (11)». This affirmation seems to be counterintuitive with the data presented in the paper and highlighted in the summary: “our results showed that in response to bacteria, thermogenic activity promoted a discrete and local anti-inflammatory environment in white adipose tissue characterized by the increase of the IL-1RA secretion”. Clarification is required on this matter.

• In Figure 1 and Figure 2, the authors presented an adjusted rectal temperature. However, in the material and method section the rectal temperature measurement and the adjustment are not mentioned. An explanation is required on this matter.

Major comments:

• In Figure 1, effects of CL316.243 a well-known browning inducer and brown adipocyte activator are assessed by measurement of temperature, weight loss, and lipid droplets size. This characterization could be more complete. For example, utilization of qPCR targeting characteristic beige and brown adipocytes genes (in particular the uncoupling protein UCP1) could provide a quantitate evaluation of browning/brown fat activation. In addition to the qPCRs, western blot or imaging (similar to those performed in Figure 5) could assess protein levels. Finally, because CL316.243 is an adrenergic agonist, it is lipolytic inducer. Then the size of lipid droplets is not the ideal parameter to evaluate the activation/recruitment of beige and brown adipocytes.

• In Figure 2B, the initial bacteremia seems to be lower (about 25%) in mice treated with CL316.243 compare to control mice although not significant due to the standard deviations. Thus, if we consider the slope of the graph (which corresponds to the bacteremia reduction rate), it is lower in the treated mice than in the untreated mice. So, should the authors reconsider the conclusion based on the slope? To avoid any doubt, it would be good to have an identical initial bacteremia, and if possible, to increase the number of measurements.

• In Figure 5: the authors use histology to characterize the infiltration of immune cells into subcutaneous adipose tissue. Although informative, these techniques do not provide a global overview of the whole tissue. I think that cytometry utilization after isolation of stromal vascular fraction could provide quantitative data about the different immune populations present in the tissue. In addition, it would be interesting to analyze this immune infiltration after infection in kinetic and in brown adipose tissue.

• Finally, several questions remain unanswered, and it would be interesting to answer or discuss them. Firstly, the role of beige and brown adipocytes has been studied here following a global infection, induced by the systemic injection of E.coli (strain UTI89, 2x107 CFU/ mouse). Do beige and brown adipocytes have an effect in response to local infection, directly in the tissue itself or in adjacent tissues? Secondly, the authors monitored mice during 48 hours after injection. This timing make sense because it is the time necessary for the organism to reduce the bacteremia to 0 (Cf figure 2B). However, this timing being pretty short, it would be interesting to determine the role of these beige and brown adipocytes following a long-term infection. Thirdly, a single dose of E.coli has been used in this publication, it could be interesting to determine if, face to a larger or even lethal infection, brown and beige adipocytes functions would not be exacerbated and thus more obvious. A survival curve in response to a lethal infection perform in CL316.243 treated or untreated mice could be an interesting addition.

Conclusion:

Based on the presented work I think that, although convincing, the data are not, at this stage, complete enough to be published in PLOS One.

Reviewer #2: In this paper, Munro et al investigate the inflammatory response of adipose tissues to bacterial infection in mice that were pre-treated with a B3-adrenergic agonist, inducing activation of brown adipose tissue (BAT) and browning of white adipose tissue (WAT). BAT activation and/or WAT browning were previously shown to prevent or alleviate metabolic disorders related to obesity such as insulin resistance and cardiovascular diseases. Interestingly, as mentioned by the authors, obesity has been associated with endotoxaemia consecutive to alterations in the intestinal barrier. Endotoxaemia results in local adipose tissue inflammation and dysfunction, thereby contributing to obesity-induced metabolic disorders. Hence, it is of importance to determine whether BAT activation and/or WAT browning may improve metabolic profile through modulation of the inflammatory response of adipose tissues.

The findings reported here show no alteration of the inflammatory response in adipose tissues of mice treated with the B3-agonist compared to vehicle, suggesting that activation of adipose thermogenesis does not play a major role in acute infection. Noteworthy, this paper excludes a role for leptin in induction of fever as leptin secretion decreases while body temperature increases in both non-treated and B3-agonist treated mice.

Although the results mostly show no differences between the experimental groups, this work is of interest as it disproves the hypothesis of a role for BAT activation/WAT browning in the response to acute systemic infection. Importantly, Munro et al measured inflammatory cytokines actually secreted by different adipose depots and not only mRNA levels (as usually done in most papers), strengthening the physiological relevance of this study.

Major concerns:

1/ The efficiency of B3-agonist treatment is only assessed by the morphological changes observed in adipose tissues (Figure 1E). As browning is not a homogeneous process within one adipose depot, the authors have to provide molecular data showing upregulation of classical browning markers and thermogenic genes in different adipose depots studied.

2/ The absence of significant differences in cytokine production by adipose tissues between NaCl- and CL-treated animals could be partly due to the heterogeneity of the values obtained within each group. In line with comment 1, correlations between molecular markers of BAT activation or WAT browning and production of cytokines by the different depots could be informative.

3/ Figure 2B: although there are no statistical differences between NaCl- and CL-treated mice, bacteremia seems to be different and/or heterogeneous, especially at the first timepoint. Such difference has to be commented, as well as its potential effects on the different inflammatory parameters measured subsequently.

4/ Adipose tissues comprise numerous cell types including adipocytes and immune cells, both of which could contribute to cytokine secretion. On one hand, cytokine production was measured from BAT and WAT samples; therefore, it cannot be excluded that the absence of differences between NaCl- and CL-treated animals observed at the tissue level actually reflects the contribution of immune cells, which could mask a differential response of the adipocytes at the cellular level. On the second hand, WAT browning has been associated with changes M1/M2 macrophages relative amount, which is expected to result in differences in cytokine production upon B3-agonist treatment. These two points should be discussed.

Minor concerns:

1/ In the materials and methods, it is not clear whether the CL treatment was stopped when bacteria were injected or whether it was prolonged for 2 more days until mice were sacrificed.

2/ Page 8, line 3: authors mention an increase in fatty acid oxidation inferred from a decrease in plasma TG. This is over-interpretation and has to be toned down.

3/ Pages 12-13: results show higher IL-1RA secretion by WAT after CL treatment, not by brite adipocytes (as WAT is not only made of adipocytes).

4/ The whole manuscript has to be re-checked for typos (CL 312,243/CL 316,243,...) and proper English grammar (shown/showed,…).

Reviewer #3: In this study, the authors evaluate the impact of thermogenesis on infectious response during bacteremia in mice. After one week of treatment with a β3-adrenergic receptor agonist (CL316,243), they induce acute bacterial infection with E. coli and they show an IL-1RA-mediated immunomodulatory activity of thermogenic adipocytes. These results are in accordance with a previous study in which the authors demonstrated an IL-1RA anti-inflammatory response to LPS after activation of brown adipose tissue. However, in the present study, the authors do not show an alteration in adipose tissue and a limited pro- and anti- inflammatory cytokines production. These discrepancies between LPS and bacteria infection could be explained by the hypothesis of “an innate immune function of the adipose tissue in chronic rather than in acute infections”. Despite of the limited impact of this work because of some negative results, the use of bacterial infection is more physiological relevant to trigger sepsis than the use of LPS and this study open the way to further works. It highlights the importance to perform a chronic rather than an acute infection to evaluate the impact of adipose tissue on bacterial infection in a physiological context.

Study design is well described and results are clearly analysed and discussed but there are minor concerns regarding some mistake and points that should be clarified:

1/ There are some mistakes in the legend of the figure, notably regarding the number of samples:

• Figure 1: n = 12 -16 or 8 (A) or 6- 8 (B-D)

• Figure 2: n = 6 (mice) or 8 -4 (scWAT explants). The authors forgot to mention n=8 (plasma)

• Figure3: There is a mistake in the legend: “IL-2 levels were assessed in the plasma (A) or in the media of iBAT (AB) and scWAT (BC)”. n = 6 -8 (plasma) or 8 - 4(explants).

• Figure 4: n = 6 (plasma) or 8 -4 (explant).

• Figure 5: The authors have to mention the number of experiments represented by these pictures.

• Figure 6: n = 6- 8 (plasma) or 8 or 4 (explant).

More generally, can the authors explain why the “n” is not the same between plasmas and explants analysis?

2/ Introduction, page 3: there is a mistake, the authors mention that “mice displayed a higher pro-inflammatory response” instead of “anti-inflammatory response”.

3/ Fig 2A: Unlike the authors say that “the fever in response to bacteria is equivalent between NaCl- and CL316,243-treated mice”, this must be confirmed. As the rectal temperature is different before infection, the authors should normalize all the values to the initial value and to comment the increase induced by the infection. It seems evident that during the first 4 hours post infection, the increase is higher in the NaCl group than in the other one.

4/ Fig 3B: The decrease on IL-6, IL-12, KC/GRO seems due to the fact that only 4 mice have been analysed. If we compare with the analyses in plasma, we can observe that for these same cytokines, we distinguish a heterogeneity between two groups of 4 mice. Therefore, if the authors have selected the 4 mice that was lower in plasma cytokines to analyse iBAT, they highlight a decrease which is not very right for all the group samples. It seems that we can observe the same profile in scWAT even if the decrease is no significant. Can the authors explain this point?

The 3.3 and 3.4 title are similar, is-it a mistake?

6. PLOS authors have the option to publish the peer review history of their article (what does this mean?). If published, this will include your full peer review and any attached files.

Reviewer #1: No

Reviewer #2: No

Reviewer #3: **Yes: **Saint-Laurent Celine

---

## [Author Response · Author response to Decision Letter 0]

26 Jul 2021

We want to deeply acknowledge the editor to have accept to manage the editorial process of our article and the three reviewers who have taken time to read our article and for their highly constructive and interesting comments about our work. We hope to have respond to all their interrogations and to propose an improved version of our article.

Reviewer #1: 

Limitations:

The models used are unfortunately limited, activation of brown adipocytes and recruitment of beige adipocytes is limited to utilization of a single adrenergic agonist and the infection is limited to the utilization of a single dose of a single bacterial strain. In addition to CL316.243 utilization, authors could expose mice to cold which is the main brown/beige fat activator. In addition to this gain of function, a loss of function approach based on utilization of knock out mice (UCP1-KO for example) mice could have been considered.

We agree with the reviewer about the limitations of our study. We have added in the discussion some words to highlight these especially for cold (p11), Ucp1-ko (p12) and complete the last paragraph by adding “sepsis” (last paragraph).

Minor comments:

• In the second paragraph of the introduction the authors discuss a different response of beige and brown adipocytes after in vitro stimulation with LPS: «Interestingly, it has been shown that brown and white adipocytes respond differently to in vitro LPS stimulation and mediate different inflammatory responses (10)». A sentence explaining what the differences are would be appreciated.

We have added a sentence to explain these results (p3).

• In the second paragraph of the introduction, it is mentioned that after activation of brown and recruitment of beige adipocytes, mice infected with LPS exhibit a higher pro-inflammatory response characterized by increased secretion of IL-1RA «We recently demonstrated that after recruitment and activation of brown and brite adipocytes, mice displayed a higher pro-inflammatory response when they are exposed to lipopolysaccharides derived from E. coli bacteria (11). Especially, these mice secreted high level of interleukin-1 receptor antagonist (IL-1RA) known to counteract interleukin 1β (IL-1β) inflammatory action (11)». This affirmation seems to be counterintuitive with the data presented in the paper and highlighted in the summary: “our results showed that in response to bacteria, thermogenic activity promoted a discrete and local anti-inflammatory environment in white adipose tissue characterized by the increase of the IL-1RA secretion”. Clarification is required on this matter.

We thank the reviewer and we apologize for this mistake which disturb our message. We have corrected “pro-inflammatory” for “anti-inflammatory”. 

• In Figure 1 and Figure 2, the authors presented an adjusted rectal temperature. However, in the material and method section the rectal temperature measurement and the adjustment are not mentioned. An explanation is required on this matter.

We have deeply completed the animal analysis part of the Mat Meth as requested by the reviewer. 

Major comments:

• In Figure 1, effects of CL316.243 a well-known browning inducer and brown adipocyte activator are assessed by measurement of temperature, weight loss, and lipid droplets size. This characterization could be more complete. For example, utilization of qPCR targeting characteristic beige and brown adipocytes genes (in particular the uncoupling protein UCP1) could provide a quantitate evaluation of browning/brown fat activation. In addition to the qPCRs, western blot or imaging (similar to those performed in Figure 5) could assess protein levels. Finally, because CL316.243 is an adrenergic agonist, it is lipolytic inducer. Then the size of lipid droplets is not the ideal parameter to evaluate the activation/recruitment of beige and brown adipocytes.

We have included accordingly to reviewer’s comment a qPCR analysis of brown (Ucp1, perilipin 5) and general adipocytes (perilipin 1, adiponectin) marker expressions in adipose tissue. These are displayed in the new figure 2 and demonstrated the activation of brown adipose tissue and the recruitment of brite adipocytes within white adipose tissue. We have modified the main text accordingly.

• In Figure 2B, the initial bacteremia seems to be lower (about 25%) in mice treated with CL316.243 compare to control mice although not significant due to the standard deviations. Thus, if we consider the slope of the graph (which corresponds to the bacteremia reduction rate), it is lower in the treated mice than in the untreated mice. So, should the authors reconsider the conclusion based on the slope? To avoid any doubt, it would be good to have an identical initial bacteremia, and if possible, to increase the number of measurements.

Our results of bacteremia demonstrated a lowered but unsignificant cfu/mouse in plasma of mice treated with CL316,243 4 hours and, in a some extend, 24 hours post-infection. This result is not representative of a better clearance of bacteria by mice treated by CL316,243 compared to untreated mice. To clarify this result, we have modified the graph for a more classic representation including the 0 hour point corresponding to the original injected quantity of bacteria by mouse.

These results are consistent with the previous results obtained by our team using the same experimental model (Diabate M et al. (2015) Escherichia coli α-Hemolysin Counteracts the Anti-Virulence Innate Immune Response Triggered by the Rho GTPase Activating Toxin CNF1 during Bacteremia. PLoS Pathog 11(3): e1004732. doi:10.1371/journal.ppat.1004732 ; Dufies O et al. (2021) Escherichia coli Rho GTPase-activating toxin CNF1 mediates NLRP3 inflammasome activation via p21-activated kinases-1/2 during bacteraemia in mice. Nature Microbiology 6: 401-412. doi:10.1038/s41564-020-00832-5).

• In Figure 5: the authors use histology to characterize the infiltration of immune cells into subcutaneous adipose tissue. Although informative, these techniques do not provide a global overview of the whole tissue. I think that cytometry utilization after isolation of stromal vascular fraction could provide quantitative data about the different immune populations present in the tissue. In addition, it would be interesting to analyze this immune infiltration after infection in kinetic and in brown adipose tissue.

To complete our histological approaches, we have analysed mRNA expression of B lymphocyte (Cd19), T lymphocyte (Tcrβ) and macrophages (Cd11b) specific markers. In addition, we have analysed expression of Mrc-1, a marker of M2 macrophage. The results are displayed in the new figure 7 and shown that only Cd11b increase with bacterial infection, and no markers are affected by CL316,243. These demonstrated as expected that only monocytes and inflammatory macrophages have infiltrated adipose tissue of mice in response to bacterial infection. Indeed, lymphocyte T and B are secondarily involved in immune reaction against bacteria. We have modified the main text accordingly.

• Finally, several questions remain unanswered, and it would be interesting to answer or discuss them. Firstly, the role of beige and brown adipocytes has been studied here following a global infection, induced by the systemic injection of E.coli (strain UTI89, 2x107 CFU/ mouse). Do beige and brown adipocytes have an effect in response to local infection, directly in the tissue itself or in adjacent tissues? 

To fully answer the reviewer question, we have performed qPCR in order to detect the presence of bacteria within adipose tissue at the end of experiment. We used 2 couple of primers highly specific to the strain we used targeting 2 major toxins of the UTI89 strain CNF1 and HlyA. The CNF1 toxin bacterial RNA has been detected only in one mouse (NaCl-bacteria group) and HLYA toxin in none of them. This indicates that almost no or very low quantity of bacteria are present in the adipose tissue 48 hours after infection, consistent with what we observed for the bacterial load in the blood.

It will be very interesting to perform another study with a sustained infection to overload the immune system and increase the bacterial load in organs, or as suggested by the reviewer by directly injected bacteria within the adipose tissue. We have introduced this point in the discussion section as requested by the reviewer.

Secondly, the authors monitored mice during 48 hours after injection. This timing make sense because it is the time necessary for the organism to reduce the bacteremia to 0 (Cf figure 2B). However, this timing being pretty short, it would be interesting to determine the role of these beige and brown adipocytes following a long-term infection. 

We have introduced this point in the discussion section as requested by the reviewer.

Thirdly, a single dose of E.coli has been used in this publication, it could be interesting to determine if, face to a larger or even lethal infection, brown and beige adipocytes functions would not be exacerbated and thus more obvious. A survival curve in response to a lethal infection perform in CL316.243 treated or untreated mice could be an interesting addition.

We are convinced that this study using a model of sepsis will be highly interesting as we discussed it, but in this work, we wanted to decipher the role of brown and brite adipocytes especially in acute infection. We have introduced this point in the discussion section as requested by the reviewer.

 

Reviewer #2: 

Major concerns:

1/ The efficiency of B3-agonist treatment is only assessed by the morphological changes observed in adipose tissues (Figure 1E). As browning is not a homogeneous process within one adipose depot, the authors have to provide molecular data showing upregulation of classical browning markers and thermogenic genes in different adipose depots studied.

We have included accordingly to reviewer’s comment a qPCR analysis of brown (Ucp1, perilipin 5) and general adipocytes (perilipin 1, adiponectin) marker expressions in adipose tissue. These are displayed in the new figure 2 and demonstrated the activation of brown adipose tissue and the recruitment of brite adipocytes within white adipose tissue. We have modified the main text accordingly.

2/ The absence of significant differences in cytokine production by adipose tissues between NaCl- and CL-treated animals could be partly due to the heterogeneity of the values obtained within each group. In line with comment 1, correlations between molecular markers of BAT activation or WAT browning and production of cytokines by the different depots could be informative.

Accordingly with the reviewer’s comment, we have correlated Ucp1 mRNA levels and cytokines production in explants of scWAT and of iBAT of the same mice. As shown below for the reviewer only, no clear correlation appeared between Ucp1 levels and cytokines production in scWAT excepted IL1RA already displayed significatively different in Fig.8. For iBAT, IFNg and IL-1b we found an anticorrelation with Ucp1 level as already shown in Fig.3 and Fig.8 respectively. IL-2 production displayed a tendency to be anti-correlated to Ucp1 levels but unsignificantly. 

As this approach did not allow to clearly display new information, we have decided to not include it in our article.

3/ Figure 2B: although there are no statistical differences between NaCl- and CL-treated mice, bacteremia seems to be different and/or heterogeneous, especially at the first timepoint. Such difference has to be commented, as well as its potential effects on the different inflammatory parameters measured subsequently.

Indeed, our results of bacteremia demonstrated a lowered but unsignificant cfu/mouse in plasma of mice treated with CL316,243 4 hours and, in a some extend, 24 hours post-infection. This result is not representative of a better clearance of bacteria by mice treated by CL316,243 compared to untreated mice. To clarify this result, we have modified the graph for a more classic representation with a log10 scale and including the 0 hour point corresponding to the original injected quantity of bacteria by mouse.

These results are consistent with the previous results obtained by our team using the same experimental model (Diabate M et al. (2015) Escherichia coli α-Hemolysin Counteracts the Anti-Virulence Innate Immune Response Triggered by the Rho GTPase Activating Toxin CNF1 during Bacteremia. PLoS Pathog 11(3): e1004732. doi:10.1371/journal.ppat.1004732 ; Dufies O et al. (2021) Escherichia coli Rho GTPase-activating toxin CNF1 mediates NLRP3 inflammasome activation via p21-activated kinases-1/2 during bacteraemia in mice. Nature Microbiology 6: 401-412. doi:10.1038/s41564-020-00832-5).

4/ Adipose tissues comprise numerous cell types including adipocytes and immune cells, both of which could contribute to cytokine secretion. On one hand, cytokine production was measured from BAT and WAT samples; therefore, it cannot be excluded that the absence of differences between NaCl- and CL-treated animals observed at the tissue level actually reflects the contribution of immune cells, which could mask a differential response of the adipocytes at the cellular level. 

To complete our histological approaches, we have analysed mRNA expression of B lymphocyte (Cd19), T lymphocyte (Tcrβ) and macrophages (Cd11b) specific markers. In addition, we have analysed expression of Mrc-1, a marker of M2 macrophage. The results are displayed in the new figure 7 and show that only Cd11b increase with bacterial infection, and no markers are affected by CL316,243. These demonstrated as expected that only monocytes and inflammatory macrophages have infiltrated adipose tissue of mice in response to bacterial infection. 

We agree that our results displayed whole cytokine production within the tissue and that we are not able to decipher the participation of immune cells and adipocytes to these productions which are not the objective of our work. All along the manuscript we discuss about tissue production and phenotype, without discrimination in the involvement of the different cells constituting adipose tissue. Nevertheless, as we found an increase in IL-1RA production without modification of Mrc-1 expression, we can suppose that IL-1RA is mainly produced by adipocytes as shown in our previous work (Munro et al. AJP 2020). To do not complexify our message we prefer to do not discuss about these points in our manuscript.

On the second hand, WAT browning has been associated with changes M1/M2 macrophages relative amount, which is expected to result in differences in cytokine production upon B3-agonist treatment. These two points should be discussed.

As discussed just before, Cd11b and Mrc-1 mRNA levels were unaffected by CL316,243, suggesting no major modification in M1 and M2 adipose tissue content. We have introduced this point in the discussion section as requested by the reviewer.

Minor concerns:

1/ In the materials and methods, it is not clear whether the CL treatment was stopped when bacteria were injected or whether it was prolonged for 2 more days until mice were sacrificed.

We have clarified this point of the mat met.

2/ Page 8, line 3: authors mention an increase in fatty acid oxidation inferred from a decrease in plasma TG. This is over-interpretation and has to be toned down.

The sentence has been modified to tone down the conclusion.

3/ Pages 12-13: results show higher IL-1RA secretion by WAT after CL treatment, not by brite adipocytes (as WAT is not only made of adipocytes).

Modified accordingly by “WAT containing brite adipocytes…”.

4/ The whole manuscript has to be re-checked for typos (CL 312,243/CL 316,243,...) and proper English grammar (shown/showed,…).

We have rechecked all the manuscript to fix typo errors. 

Reviewer #3: 

1/ There are some mistakes in the legend of the figure, notably regarding the number of samples:

We really apologize for these numerous errors in the number of n. We have fixed all errors.

More generally, can the authors explain why the “n” is not the same between plasmas and explants analysis?

Each group includes 8 mice. We have measured general parameters and plasma in all of these (n=8). Then, we have used for half of the mice the right inguinal scWAT and the right lobe of iBAT for histology (n=4) and the lefts for molecular analysis. Finally, for the other half of the mice the right inguinal scWAT and the right lobe of iBAT have been used for explant secretion analysis (n=4) and the lefts for molecular analysis. To be sure to have representative analysis of the tissue, we currently exclude to cut any fat pad in different pieces as they are heterogeneous, especially for browning of scWAT.

We have modified the mat meth to highlight this point.

2/ Introduction, page 3: there is a mistake, the authors mention that “mice displayed a higher pro-inflammatory response” instead of “anti-inflammatory response”.

We thank the reviewer to pinpoint this error. We fixed-it.

3/ Fig 2A: Unlike the authors say that “the fever in response to bacteria is equivalent between NaCl- and CL316,243-treated mice”, this must be confirmed. As the rectal temperature is different before infection, the authors should normalize all the values to the initial value and to comment the increase induced by the infection. It seems evident that during the first 4 hours post infection, the increase is higher in the NaCl group than in the other one.

We agree with the reviewer that only untreated mice displayed an increase in rectal temperature after infection. Nevertheless, pyretic response or fever is the same between the two groups of mice after infection. As discussed in p11, this result indicated that fever and thermogenesis are not additive. 

4/ Fig 3B: The decrease on IL-6, IL-12, KC/GRO seems due to the fact that only 4 mice have been analysed. If we compare with the analyses in plasma, we can observe that for these same cytokines, we distinguish a heterogeneity between two groups of 4 mice. Therefore, if the authors have selected the 4 mice that was lower in plasma cytokines to analyse iBAT, they highlight a decrease which is not very right for all the group samples. It seems that we can observe the same profile in scWAT even if the decrease is no significant. Can the authors explain this point?

As mentioned by the reviewer, plasma levels of various cytokines highlight two groups of mice. We have hypothesized that this can reflect a delayed response to bacterial infection. Unfortunately, we have not been able to correlate this to the bacterial load in the blood of mice. Nevertheless, these plasma levels did not influence adipose tissue cytokines secretion. Indeed, first we found for several of these an inverse profile as for IL-6 which increased in plasma after infection and decreased in tissue. Then, we fortunately analysed mice of the two “plasma cytokine profiles” in adipose tissue explant secretion assay. For example, two mice displaying high quantity and 2 mice displaying low quantity of IL-6 in plasma after infection have been used for explant secretion analysis. This is the case for untreated and treated mice. 

The 3.3 and 3.4 title are similar, is-it a mistake?

We thank the reviewer and we have modified the 3.4 title.

---

## [Editor Report · Decision Letter 1]

16 Aug 2021

Impact of thermogenesis induced by chronic β3-adrenergic receptor agonist treatment on inflammatory and infectious response during bacteremia in mice

PONE-D-21-05608R1

Dear Dr. Pisani,

We’re pleased to inform you that your manuscript has been judged scientifically suitable for publication and will be formally accepted for publication once it meets all outstanding technical requirements.

Kind regards,

Fulvio D'Acquisto, PhD

Academic Editor

PLOS ONE
---

## [Editor Report · Acceptance letter]

17 Aug 2021

PONE-D-21-05608R1 

Impact of thermogenesis induced by chronic β3-adrenergic receptor agonist treatment on inflammatory and infectious response during bacteremia in mice 

Dear Dr. Pisani:

I'm pleased to inform you that your manuscript has been deemed suitable for publication in PLOS ONE. Congratulations! Your manuscript is now with our production department. 

Kind regards, 

on behalf of

Professor Fulvio D'Acquisto 

Academic Editor

PLOS ONE